# Which resources help young people to prevent and overcome mental distress in deprived urban areas in Latin America? A protocol for a prospective cohort study

Stefan Priebe,[1] Catherine Fung  ,[1] Luis Ignacio Brusco,[2] Fernando Carbonetti,[2] Carlos Gómez-Restrepo,[3,4] Miguel Uribe,[4] Francisco Diez-Canseco,[5] Melanie Smuk,[6] Nicola Holt,[7] James B. Kirkbride  ,[8] Ricardo Araya,[9] Craig Morgan,[9] Sandra Eldridge,[10] Paul Heritage,[11] Victoria Bird[1]

For numbered affiliations see end of article.

**Correspondence to**
Catherine Fung;
c.fung@qmul.ac.uk

## ABSTRACT

**Introduction** Improving the mental health of young people is a global public health priority. In Latin America, young people living in deprived urban areas face various risk factors for mental distress. However, most either do not develop mental distress in the form of depression and anxiety, or recover within a year without treatment from mental health services. This research programme seeks to identify the personal and social resources that help young people to prevent and recover from mental distress.

**Methods and analysis** A cross-sectional study will compare personal and social resources used by 1020 young people (aged 15–16 and 20–24 years) with symptoms of depression and/or anxiety and 1020 without. A longitudinal cohort study will follow-up young people with mental distress after 6 months and 1 year and compare resource use in those who do and do not recover. An experience sampling method study will intensively assess activities, experiences and mental distress in subgroups over short time periods. Finally, we will develop case studies highlighting existing initiatives that effectively support young people to prevent and recover from mental distress. The analysis will assess differences between young people with and without distress at baseline using t-tests and $\chi^2$ tests. Within the groups with mental distress, multivariate logistic regression analyses using a random effects model will assess the relationship between predictor variables and recovery.

**Ethics and dissemination** Ethics approvals are received from Ethics Committee in Biomedical Research, Faculty of Medicine, University of Buenos Aires; Faculty of Medicine-Research and Ethics Committee of the Pontificia Universidad Javeriana, Bogotá; Institutional Ethics Committee of Research of the Universidad Peruana Cayetano Heredia and Queen Mary Ethics of Research Committee. Dissemination will include arts-based methods and target different audiences such as national stakeholders, researchers from different disciplines and the general public.

**Trial registration number** ISRCTN72241383.

## Strengths and limitations of this study

► The large sample size will allow both comparisons of young people with and without distress and the exploration of factors predicting recovery in two relevant age groups.

► The inclusion of three diverse Latin American countries provides the possibility to explore commonalities and differences across different contexts.

► Inclusion of experience sampling methodology enables exploration of resource use in short-term recovery (within hours/days) and comparison with long-term recovery (over 1 year).

► Traditional methods of collecting data, that is, completing questionnaires and in-depth interviews may not be the most effective ways of engaging young people in research.

► The ongoing pandemic requires flexibility for online and offline data collection, which may increase the variance of findings and reduce the statistical power to identify differences between groups and predictor variables for outcomes.

## INTRODUCTION

Adolescence and young adulthood are critical periods of social, behavioural and psychological maturation and change. These periods create opportunities for the development of lifelong well-being and resilience in the face of future adversities. However, they are also associated with a significant risk for developing mental ill health, and estimates suggest that 75% of all mental disorders begin by the age of 24.[1] WHO has identified improving mental health among young people as a key priority required to promote sustained economic and social development.[2] Poor mental health in the form of depression

and anxiety is associated with high levels of distress and disability, future physical and psychiatric morbidity and educational and social impairment.[3] Furthermore, over half of all adolescent suicides are attributable to depression,[4] making it the leading cause of mortality for this age group.[5]

Urban regions, predominantly large cities, account for 81% of the population in Latin America, making it one of the most urbanised regions in the world.[6] Within these environments, people are frequently exposed to various risk factors for poor mental health. These include poverty, social fragmentation, poor education, poor housing, low employment rates, gang warfare, victimisation, violence and widespread substance misuse.[7–10]

Depression and anxiety during adolescence and youth are a particular concern within low-income and middle-income countries (LMICs), where the burden of common mental disorders is greatest.[2 11 12] This includes Latin America, where young people represent one-quarter of the population. Estimated levels of depression and/or anxiety for adolescents within the region range from 17% in Colombia[13] to 26% in Argentina.[14] Those exposed to adversities such as violent conflicts, internal displacement and poverty are at an even greater risk.[15] However, despite all the risk factors, the majority of young people do not suffer from depression and/or anxiety.

Due to the scarcity of financial and human resources,[16] young people in deprived areas in Latin America rarely receive formal treatment when experiencing mental distress. Yet, evidence suggests that 50%–60% experience symptomatic recovery within 1 year.[17 18] This raises the question as to which resources individuals in these contexts mobilise to prevent and overcome mental distress and build mental health resilience.

Resilience has been defined as withstanding or overcoming adversity, trauma and stress and rebounding back from distress.[19] Thus, it covers the two processes: preventing mental distress in the face of adversity and recovering from distress if and when it develops. A systematic review into the concept of resilience within mental health classified its characteristics into two components—personal and social resources.[20] Resources encompass a wide range of strengths, assets, materials, sources of information or help and means of support available to the individual. They are commonly mobilised through and in activities. These activities may be individual and reflect the health behaviours or the skills and abilities of the person[21] (personal resources); or group based and at the community or societal level, such as captured in the concepts of social capital, social connectedness, social identity or social networks (social resources).[22] Resources may be specific to culture, context and locality (eg, local music group).

Previous research has focused mainly on risk factors for developing mental disorders. However, complete primary prevention of mental distress among young people, particularly in adverse urban environments, is unrealistic. This programme, therefore, aims to identify which personal and social resources young people use and which ones help them to overcome mental distress, reflecting the concepts of secondary and tertiary prevention[23] as well as recovery. This crucial step may lead to future interventions to reduce the burden of mental distress in young people living in deprived urban neighbourhoods in Latin America and other LMICs and help them to maintain good mental health throughout the rest of their lives.

## Programme collaborators

This research programme is coordinated by the Unit for Social and Community Psychiatry (WHO Collaborating Centre for Mental Health Service Development) at Queen Mary University of London, and conducted in collaboration with researchers from Universidad de Buenos Aires, Pontificia Universidad Javeriana (Bogotá), Universidad Peruana Cayetano Heredia (Lima), University College London and King's College London. The research activities will take place in deprived areas of three large Latin American cities: Buenos Aires (Argentina), Bogotá (Colombia) and Lima (Peru).

A unique aspect of this programme is the partnership with well-established community-based arts organisations in each city: Fundación Crear Vale la Pena (Buenos Aires), Fundación Batuta and Familia Ayara (Bogotá), and Teatro La Plaza (Lima). Through specific art forms and creativity, these organisations work to support adolescents and young people living in vulnerable neighbourhoods to improve their quality of life, strengthen socioemotional skills, and express and reflect on social realities and problems affecting young people. Arts-based activities are well documented in their power to engage young people in research and to explore sensitive topics such as mental health, facilitate wider discussions and generate new knowledge.[24 25] The purpose of these collaborations is therefore to work through the arts to involve young people in this programme and its discourse about how to prevent and overcome mental distress; to explore the use of arts-based methodologies to understand resource use in young people; and to use arts practices to disseminate the findings to a wider, non-academic audience.

## Objectives

The overall aim is to identify which resources help young people living in deprived urban environments in Latin America prevent and recover from depression and/or anxiety.

The aim will be addressed via specific objectives:
1. Determine whether resource use differs between adolescents and young adults with and without depression and/or anxiety.
2. Identify which resources help adolescents and young adults recover from depression and/or anxiety over a 1-year period.
3. Explore existing approaches and resource-oriented interventions that are effective in preventing depression and/or anxiety in adolescents and young adults or in supporting them to recover.

**Figure 1: Study design**

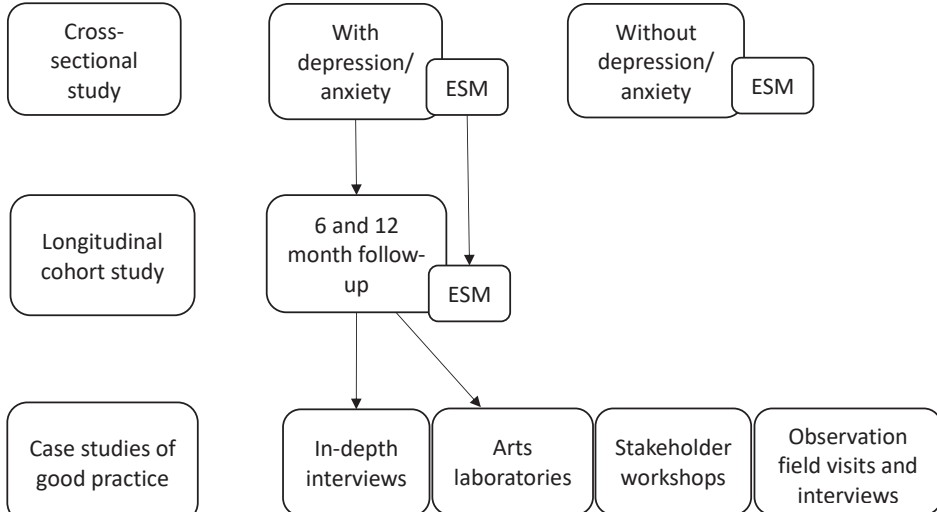

**Figure 1** Study design. ESM, experience sampling methodology.

## METHODS AND ANALYSIS
### Study design
The programme consists of several observational studies, using quantitative and qualitative methods (see figure 1 for an overview of the study design). The methods have been finalised in a preparatory phase with workshops, focus groups and pilot interviews with young people in each of the three cities. The research activities are planned with procedures in place for both online and face-to-face data collection depending on the current local situation of the COVID-19 pandemic and government restrictions.

### Settings
This study will take place in Buenos Aires (Argentina), Bogotá (Colombia) and Lima (Peru). In each city, a reproducible strategy was developed to sample target participants from the most deprived half of districts in each city. This involved reviewing the main indices used to collect routine statistics on the population in each setting, and using those indices which maximised comparability between settings as the basis for participant sampling. In Lima (Peru) and Bogotá (Colombia), we used the United Nations Development Programme's Human Development Index (HDI)[26] to assess the proportion of households in each district achieving basic living standards according to four criteria on life expectancy for health, expected years of schooling, mean years of schooling for education and gross national income per capita. In Buenos Aires (Argentina), where district-level statistics on the HDI were unavailable, we used the Unsatisfied Basic Needs Index[27] (known as 'NBI') to assess the proportion of households in each district experiencing unmet needs in at least one of five domains (housing, sanitary conditions, overcrowding, school attendance, subsistence capacity). In each setting, we ranked districts according to HDI/NBI levels and selected districts in the bottom 50% according to these scores. At district level, the HDI

correlated strongly with the proportion of households in the highest SES (socioeconomic status) quintile in Lima ($\rho$=0.84; N=49) and the proportion of households in multidimensional poverty in Bogotá ($\rho$=0.77; N=20).

Within these communities, recruitment of the younger age group will be mostly from schools and of the older group mainly from primary care centres. Other recruitment centres might include local arts centres, youth organisations and other community and educational organisations within defined geographical areas.

### Participants
Study participants predominantly fall into two age groups: adolescents aged 15–16 years old and young adults aged 20–24 years old, with and without mental distress, living in the defined geographical areas (see above), with capacity to provide informed consent or assent alongside parental informed consent. Exclusion criteria are any severe mental illness (psychosis, bipolar disorder, schizophrenia), cognitive impairment and illiteracy (where participants are required to self-administer questionnaires).

### Cross-sectional and longitudinal cohort studies
At each of the three sites, we will recruit 340 young people (15–16 years and 20–24 years) with mental distress and 340 young people without. Using a dimensional approach—rather than diagnostic categories—mental distress will be defined by a score of greater than nine on either the Patient Health Questionnaire-8 (PHQ-8)[28] or General Anxiety Disorder-7 (GAD-7).[29] Both scales are self-rated. The PHQ-8 assesses symptoms of depression and the GAD-7 of anxiety. Each participant will be asked to complete the developed assessment battery.

Young people identified with mental distress will be invited to take part in the longitudinal study, which will allow comparison of resource use in those who do and

do not recover. Participants will be asked to complete a brief assessment at 6 months and the same full assessment battery at 12 months (online supplemental file 1 shows the schedule of assessments).

## Experience sampling method study

Experience sampling methodology (ESM) is a method of data collection that asks participants to report on their thoughts, feelings, behaviours and environment on multiple occasions as they go about their daily activities each day for a specific time period, for example, over 7 days.[30] ESM allows for a detailed assessment of the interaction between real-world context and phenomena that is unaffected by issues of recall.[30] Data will be collected using a mobile phone app called eMoodie,[31] which was developed specifically for young people. The ESM study will run alongside the cross-sectional and longitudinal studies sharing the same aims and with the additional aim to determine whether young people use similar personal and social resources for short-term recovery (within hours or days) as for long-term recovery (over 1 year).

The ESM study will recruit 150 participants (50 in each country) already enrolled in the cross-sectional and longitudinal studies: 90 with depression and/or anxiety and 60 without, as defined above and across the two age groups (15–16 years and 20–24 years). Participants will first attend a training session with a researcher, who will explain how to use the eMoodie app, when to complete the ESM assessments, to practise the questionnaires and explain each item in detail. Participants will be given phones, and/or data cards to access the internet if this is needed. Participants can contact the researcher with any questions about, or problems with, eMoodie or the ESM questions. The researcher will conduct a midweek phone call at an agreed time to discuss any concerns or problems they are experiencing with completing the questionnaires and motivate participants to complete as many ESM questionnaires as possible.

Young adult participants (20–24 years old) will complete the ESM assessment eight times per day over 7 days, and adolescent participants (15–16 years old) will complete the ESM assessment five times per day for school days and eight times per day for weekend days, again over 7 days. The times will be scheduled at random within set blocks of time. At the end of the 7-day data collection period, participants will be asked to answer a short questionnaire to briefly explore through open questions, their experience of the ESM study and any problems encountered.

The 90 participants defined with depression and/or anxiety, who participated in both the ESM study baseline and longitudinal study, will be contacted to complete the ESM assessments again after 12 months.

## Case studies of good practice

This work activity will support knowledge exchange and suggest practical implications. Case studies of good practice models will illustrate and explain our previous findings. Based on the longitudinal study, we will identify and describe small-scale interventions already happening within the communities (eg, adolescent well-being centres), and health and education services (eg, school-based resilience programmes), in order to assess how communities and services can effectively prevent or respond to mental distress in young people.

Arts-based laboratories will run across 6–8 weeks by the partner arts organisations in each city to explore attitudes and perceptions about mental distress in 30 participants from the longitudinal study (10 in each city) in order to reveal individual, social and contextual resilience factors and the significance of different personal and social resources in their own recovery from depression and/ or anxiety. The methodologies developed in the preparatory work for researcher participation/observation will be applied at key moments throughout the workshop programme and in the discussions with young people and the arts facilitators from each organisation. Longitudinal study participants will also have the opportunity to participate in integrated arts workshops with other young people in each arts organisation.

To identify areas of good practice, 30 longitudinal study participants will be invited to take part in in-depth interviews as soon as possible after completing the 12-month follow-up. Interviews will explore which resources individuals used and found helpful or not helpful, which resources they are aware of, and any suggestions for ways in which the community or/and services could help them with their recovery. These interviews will be used alongside a stakeholder consultation workshop to recommend areas of good practice within each city, including different types of services, initiatives, projects or approaches. This workshop will be conducted with facilitated discussion asking for visual and written descriptions of existing projects and approaches. Field visits and interviews with individuals who work at and/or use these initiatives will be conducted to aid scale-up and dissemination. Finally, a stakeholder consultation will bring together the information collected to produce the case studies of each approach.

## Measures

For the cross-sectional and longitudinal studies, data will be collected using a case report form (CRF). Each country-specific CRF contains the relevant language variations. A small number of questions will be varied to collect locally relevant data (socio-demographic characteristics, health insurances) and in-line with local regulations (sexual activities). All countries will be assessing the same key outcomes and resources: personal background and characteristics, social context, mental distress, quality of life, life events, experiences and the actual use of personal resources and social resources. Online supplemental file 1 summarises the questionnaires and the data collected at baseline, 6 and 12 months.

At baseline, sociodemographic data will be collected, including information about age, gender, living situation, current or obtained level of education, employment,

current and previous experiences of depression and anxiety, and family history of depression and anxiety. Variables that might change over a 1 year period will be collected again at the 12-month follow-up, including current main occupation and current experiences of depression and anxiety.

The ESM assessments will ask participants about their current affect, arousal state, location, company and activity (see online supplemental file 2).

## Bias

Adolescents will be primarily recruited from schools and young adults from primary care centres to ensure we recruit a varied sample of young people. Samples for both age groups will include a balance of gender. Resource use may be affected by the presence of the pandemic and restrictions to leaving homes during lockdown, but we will assess resource use as planned and collect information about lockdown in each city throughout data collection, to facilitate further data analysis as required.

## Sample size calculations

The sample size for the cross-sectional and longitudinal studies has been calculated for identifying variables that predict recovery over a 1-year period. Assuming a predictor variable (personal or social resource) exists in 10% of the population, we can detect a difference in recovery rate between 40% and 60% with 90% power and at a 5% significance level with a total sample of 762 people. Assuming a dropout rate of 25%, 1016 participants with mental distress are required at baseline. Recruiting 340 people with mental distress in each of the three countries provides a total sample of 1020.

## Patient and public involvement

The study is not a clinical study recruiting patients. Participants will be young people with and without mental distress in the deprived communities in Latin American cities. Young people in Bogotá, Buenos Aires and Lima were extensively involved in the preparation of the study. In arts-based workshops, focus groups and pilots of the assessment battery, they helped to develop appropriate methods for assessing resource use. For the arts-based workshops and the arts laboratories, we are working with our partner arts organisations, which use a variety of artistic languages that can encourage engagement in research. Focus groups with adolescents (aged 15–16 years), young adults (aged 20–24 years), and professionals (mental health professionals, educators, staff from youth organisations) explored views on resources that young people use to prevent and overcome mental distress. Pilots of the assessment battery established the feasibility and acceptability of an assessment battery, including the burden of completing this through timed pilots. The findings from the preparatory work allowed us to refine the assessment measures to reflect young people's experiences, priorities and preferences. The overall research question and design of the study were informed by the literature and expertise of the multidisciplinary research team, and patients/the public were not involved in this process.

Each research team has established a Lived Experience Advisory Panel (LEAP) composed of young people aged 15–24 years old with lived experience of depression and/or anxiety. The purpose of the LEAPs is to engage young people and ensure their voice is present and heard throughout the research, including recruitment to and conduct of the research. The LEAP panel and our partner arts organisations will be consulted about the dissemination of findings to wider audiences of young people to continue the discourse about mental health distress.

## Data analysis

Local researchers will transcribe verbatim the audio or video recordings from the in-depth interviews and will analyse using framework analysis following the stages of familiarisation; identifying a thematic framework; indexing; charting; and interpretation.[32] This will focus on common features of the identified practices and resources, which could be scaled up and more widely disseminated.

For the cross-sectional and longitudinal studies, data will be collected on paper CRFs with pseudonymised data entered onto a secure online database called REDCap (Research Electronic Data Capture), and/or participants will complete the electronic form that captures data directly onto the same database. Descriptive statistics will be reported for sociodemographic data for all participants. The numbers of individuals at each stage of the study will be reported, including the numbers potentially eligible, screened, confirmed as eligible, included, followed up and analysed.

For the cross-sectional study, differences between the groups on each of the measures will be assessed using t-tests and $\chi^2$ tests to determine whether resource use differs between young people with and without depression and/or anxiety and whether the differences are apparent for the two age groups and gender.

We understand long-term recovery as recovery over a 1-year period. At 1-year follow-up, based on the PHQ-8 and GAD-7 scores, individuals will be classified as recovered—defined as no longer screening positive for depression and/or anxiety (scoring 9 or less on either scale), or not recovered (scoring greater than 9 on either scale). The distribution and level of resources will be summarised using the median, while frequency and counts will summarise exposure to different community resources. A multivariate logistic regression analysis using a random effects model will assess the relationship between predictor variables and recovery, first for each age group, and then including the whole sample to determine whether the same resources are associated with recovery for all age groups and genders.

Data from the ESM assessments will be collected using the eMoodie mobile phone app, and then stored on a secure server. The data will be multilevel as multiple

observations are nested within participants. Short-term recovery means any recovery within the 7-day assessment period, whether it is within hours or over the full 7 days. Recovery will be assessed as a continuous variable, as improvement of any reported distress. The analysis of this will be exploratory and consider different levels of distress and improvement. Long-term recovery will use the same definition as described above. Multilevel models will assess the link between stress-related variables and negative and positive affect, controlling for potential confounding factors such as gender and socio-demographic variables. Differences between groups will be assessed, and different types of resources will be categorised and the frequency of resources within each category calculated.

## ETHICS AND DISSEMINATION

For all research activities, informed consent will be obtained from participants, or assent will be obtained from participants under 18 years old alongside informed consent from their parent or legal guardian. Each research team will provide participants with information about depression and anxiety, and organisations and resources in each city for mental health support. Procedures are in place to manage any instances of risk of harm to participants or others. Where researchers conduct face-to-face data collection, they are required to follow their institutional and/or government guidelines to protect themselves and participants from COVID-19 infection. All data will be pseudonymised, handled and stored in line with the Data Protection Act 2018, General Data Protection Regulation, and national data protection laws in each partner country. Only research team members will have access to data during the study, which will be stored on REDCap, hosted on secure servers at Queen Mary, University of London. To protect the identity of participants, they will each be assigned a participant identification number (PIN). The list linking participants' names to their PIN will be stored separately and securely. After the completion of the programme, the datasets used and/or analysed during the cross-sectional, longitudinal and ESM studies will be available in a publicly available repository. Participants' consent (or their parent/guardian's consent if under 18 years old) to share deidentified will be obtained for this purpose.

The findings will be disseminated to national stakeholders, including policy-makers, professionals in healthcare and education, charities and non-governmental organisations. Traditional routes of dissemination include publishing open access in peer-reviewed journals and presenting findings at national and international conferences. Through our partner arts organisations, we will also enable young people to disseminate findings to a wider, non-academic audience, including their own peer groups.

**Author affiliations**
[1]Unit of Social and Community Psychiatry (WHO Collaborating Centre for Mental Health Service Development), Queen Mary University of London, London, UK
[2]Department of Psychiatry and Mental Health, School of Medicine, Universidad de Buenos Aires, Buenos Aires, Argentina
[3]Department of Clinical Epidemiology and Biostatistics, Pontificia Universidad Javeriana, Bogota, Colombia
[4]Department of Psychiatry and Mental Health, Pontificia Universidad Javeriana, Bogota, Colombia
[5]CRONICAS Center of Excellence in Chronic Diseases, Universidad Peruana Cayetano Heredia, Lima, Peru
[6]Medical Statistics, London School of Hygiene & Tropical Medicine, London, UK
[7]Department of Health and Social Sciences, University of the West of England, Bristol, UK
[8]Division of Psychiatry, University College London, London, UK
[9]Health Service and Population Research, King's College London, London, UK
[10]Pragmatic Clinical Trials Unit, Queen Mary University of London, London, UK
[11]School of English and Drama, Queen Mary University of London, London, UK

**Acknowledgements** We would like to acknowledge our partner arts organisations for their ongoing contribution to the OLA programme: Fundación Crear Vale la Pena (Buenos Aires), Fundación Batuta and Familia Ayara (Bogotá), and Teatro La Plaza (Lima).

**Contributors** SP and VB led on the conception, design and organisation of the programme. CF and SP wrote the first draft of this manuscript. All authors contributed to the whole study design, commented on earlier drafts, and read and approved the manuscript. MS and SE particularly contributed to the statistical aspects of the programme, JBK to the method for identifying appropriate geographical areas, PH to the planning of all arts activities, NH to the adaptation of the Experience Sampling Method, CM to conceptual and methodological issues of the cohort study, RA to considerations of the Latin American context, and CF to organisational aspects of the programme. LIB, FC, CG-R, MU and FD-C specifically contributed to the adaptation of the design to the national contexts in Argentina, Colombia and Peru.

**Funding** This work is supported by the Medical Research Council (grant number: MR/S03580X/1).

**Competing interests** None declared.

**Patient consent for publication** Not required.

**Provenance and peer review** Not commissioned; externally peer reviewed.

**ORCID iDs**
Catherine Fung http://orcid.org/0000-0002-3220-6930
James B. Kirkbride http://orcid.org/0000-0003-3401-0824

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
