## [Reviewer comments · BMJ Open]

ARTICLE DETAILS

TITLE (PROVISIONAL)	Which resources help young people to prevent and overcome mental distress in deprived urban areas in Latin America? A protocol for a prospective cohort study
AUTHORS	Priebe, Stefan; Fung, Catherine; Brusco, Luis Ignacio; Carbonetti, Fernando; Gómez-Restrepo, Carlos; Uribe, Miguel; Diez-Canseco, Francisco; Smuk, Melanie; Holt, Nicola; Kirkbride, James; Araya, Ricardo; Morgan, Craig; Eldridge, Sandra; Heritage, Paul; Bird, Victoria

VERSION 1 – REVIEW

REVIEWER	Tyrer, Peter Imperial College London, Psychological medicine.
REVIEW RETURNED	12-Jul-2021

GENERAL COMMENTS	This is a worthwhile study and I only have one comment. The centres concerned are urban ones with a considerable amount of social deprivation. If there was a way of formalising the degree of deprivation in each area (ie.equivalent of Jarman index) it would be very helpful in interpreting the results in a global context as well as comparing differences between the centres.
--

REVIEWER	Barrera, Alvaro University of Oxford, Psychiatry
REVIEW RETURNED	29-Jul-2021

GENERAL COMMENTS	Thank you for this very interesting project with potentially significant impact on young people's wellbeing. I am sure the authors will correct for multiple comparisons. There are typos in pages 20 and 22.
---

VERSION 1 – AUTHOR RESPONSE

Reviewer: 1
Prof. Peter Tyrer, Imperial College London

Response: We have not included a common measure of deprivation as there wasn't one available at city level. However, we were able to identify the Human Development Index in Bogotá and Lima and unsatisfied basic needs in Buenos Aires as comparable indices. We have added more information to the paper regarding the strategy used to identify these indices.

Reviewer: 2

Dr. Alvaro Barrera, University of Oxford

Response: As suggested, we checked all spellings on pages 20 and 22. We took the references directly from Mendeley Reference Manager so we trust that their spelling is correct. We followed the original references also in terms of British versus American spelling.